# Polish Adaptation of the Italian Spine Youth Quality of Life Questionnaire

**DOI:** 10.3390/jcm10102081

**Published:** 2021-05-12

**Authors:** Edyta Kinel, Krzysztof Korbel, Piotr Janusz, Mateusz Kozinoga, Dariusz Czaprowski, Tomasz Kotwicki

**Affiliations:** 1Department of Rehabilitation, University of Medical Sciences in Poznan, 61-545 Poznan, Poland; 2Department of Physiotherapy, University of Medical Sciences in Poznan, 61-545 Poznan, Poland; kkorbel@ump.edu.pl (K.K.); dariusz.czaprowski@interia.pl (D.C.); 3Department of Spine Disorders and Pediatric Orthopedics, University of Medical Sciences in Poznan, 61-545 Poznan, Poland; mdpjanusz@gmail.com (P.J.); mkozinoga@hotmail.com (M.K.); kotwicki@ump.edu.pl (T.K.); 4Department of Health Sciences, Olsztyn University, Bydgoska 33, 10-243 Olsztyn, Poland

**Keywords:** idiopathic scoliosis, health-related quality of life, cultural adaptation, Italian Spine Youth Quality of Life Questionnaire

## Abstract

The study aimed to carry on the process of the cultural adaptation of the Italian Spine Youth Quality of Life Questionnaire (ISYQOL) into Polish (ISYQOL-PL). The a priori hypothesis was: the ISYQOL-PL questionnaire is reliable and appropriate for adolescents with a spinal deformity. Fifty-six adolescents (mean age 13.8 ± 1.9) with idiopathic scoliosis (AIS) with a mean Cobb angle 29.1 (±9.7) and two with Scheuermann juvenile kyphosis (SJK) with a kyphosis angle 67.5 (±17.7) degrees were enrolled. All patients had been wearing a corrective TLSO brace for an average duration of 2.3 (±1.8) years. The Institutional Review Board approved the study. The cross-cultural adaptation of the ISYQOL-PL was performed following the guidelines set up by the International Quality of Life Assessment Project. The reliability was assessed using internal consistency (the Cronbach’s alpha coefficient) and test–retest reliability (intraclass correlation coefficient ICC_2.1_, CI = 95%); moreover, floor and ceiling effects were calculated. The internal consistency was satisfactory (Cronbach’s alpha coefficient 0.8). The test–retest revealed high reliability with the value of ICC_2.1_ for the entire group 0.90, CI (0.84 to 0.94). There was neither floor nor ceiling effect for the ISYQOL-PL overall score. The ISYQOL-PL is reliable and can be used in adolescents with spinal deformity.

## 1. Introduction

The impact of the disease and the treatment applied on the quality of life of adolescents with spinal deformities, namely adolescent idiopathic scoliosis (AIS) or Scheuermann juvenile kyphosis (SJK), is considered to be multidimensional. This includes subjective perception, and changes in physical, psychological, and social well-being. Moreover, spinal deformities’ progressive character can affect adolescents’ health-related quality of life (HRQoL). Therefore, while evaluating AIS and SJK adolescents’ treatment results, the changes in the HRQoL should be taken into account [1]. A variety of HRQoL questionnaires exist, dedicated to adolescents with spinal deformities having completed their treatment, e.g., Scoliosis Research Society—(SRS-22) or Scoliosis Quality of Life Index (SQLI) [2,3]. Vasiliadis et al. [4] developed the Brace Questionnaire (BrQ), an instrument for measuring the HRQoL specifically for scoliotic adolescents who are being treated conservatively by wearing a corrective brace, and the Polish version of BrQ has been validated [5]. There are questionnaires admitted to monitoring the level of stress induced by the deformity (Bad Sobernheim Stress Questionnaire, BSSQ—Deformity) and the stress induced by the treatment with a brace (Bad Sobernheim Stress Questionnaire, BSSQ—Brace) for patients with idiopathic scoliosis [6]. However, the BSSQ questionnaires do not evaluate the overall HRQoL. Recently, to address the limitations of existing HRQoL tools, the Italian Spine Youth Quality of Life (ISYQOL) questionnaire has been developed by the Italian Scientific Spine Institute. The ISYQOL questionnaire was built using Rasch analysis. The ISYQOL questionnaire measures HRQoL in all types of spinal deformities, including surgical curves [7,8]. The ISYQOL has been cross-culturally adapted and validated in English [9] and Spanish [10], but not yet in Polish. This paper aims to carry on the cultural adaptation of the ISYQOL questionnaire into Polish (ISYQOL-PL).

## 2. Materials and Methods

The a priori hypothesis was proposed as follows: the ISYQOL-PL questionnaire version is reliable and appropriate for adolescents with a spinal deformity.

### 2.1. Study Population

Fifty-eight patients with spinal deformities (56 AIS and 2 SJK) were enrolled to assess the ISYQOL-PL questionnaire version. The sample included 52 girls and six boys. The same specialist in orthopedics treated the patients. The following criteria for inclusion were applied: (1) patients at the age of 10–18 years; (2) AIS or SJK diagnosis; (3) who have been wearing the brace for at least three months for at least 12 h/24 h (4) AIS patients with a Cobb angle ≥ 20°, SJK patients with a kyphosis angle ≥ 45°. Exclusion criteria: (1) history of spine surgery; (2) combined spinal deformities (e.g., scoliosis plus spondylolysis), (3) a history of relevant diseases, surgery, or trauma, including a positive neurologic examination.

The mean age of the patients at the time of completing the questionnaire was 13.8 (±1.9) years, the mean AIS patients’ Cobb angle was 29.1 (±9.7) degrees, and the SJK with kyphosis angle was 67.5 (±17.7) degrees. All patients had been wearing a corrective TLSO brace with an average duration of 2.3 (±1.8) years. 

Twice within a one-week interval, patients were asked to fill in the questionnaire. The time needed to answer all questions during the first attempt to complete the questionnaire was measured. All the invited adolescents selected through the inclusion criteria participated in the study. The participants were left to fill in the questionnaire alone, being in a separate space, to minimize any influence from parents or medical staff. Data from the ISYQOL-PL were collected during November and December 2019. 

The Institutional Review Board approved the study.

Before inclusion in the study, the parents and the patients awarded their informed consent. 

### 2.2. Italian Spine Youth Quality of Life (ISYQOL) Questionnaire

The ISYQOL questionnaire is based on patients’ concerns and has been shown to be particularly appropriate in AIS and SJK patients undergoing non-surgical management. The ISYQOL is a 20 items questionnaire. Each item is scored 0, 1, or 2. Items investigating the presence of spine-related problems (questions 1–4, 7–9, 11–12 and 14–20) are coded 0-1-2 (0: never; 1: sometimes; 2: often). Conversely, “items investigating the presence of positive thoughts” (question 5, 6, 10, 13) are coded 2-1-0 (2: never; 1: sometimes; 0: often). It provides a total score, with lower scores representing a higher quality of life. The ordinal ISYQOL total score is subsequently converted to an interval measure (i.e., ISYQOL measure), expressed on a 0–100% scale, where 100% indicates the highest quality of life. 

The Rasch method used in the analysis allows for a comparison of the ISYQOL result of non-brace wearers (who answer only 13 of the 20 items) with brace wearers (who complete the entire questionnaire). The questionnaire is developmentally appropriate for ages 10–18 years and designed to be self-administrated [7,8]. 

### 2.3. Adaptation Process

The process of the cross-cultural adaptation of the ISYQOL-PL was performed following the guidelines set up by the International Quality of Life Assessment (IQOLA) [11]. The implementation of this method includes the following steps: (1) forward translation, (2) back-translation and expert panel, (3) pre-testing and cognitive interviewing, (4) development of the final version. The total sample size was decided based on previous recommendations for validation studies [12].

### 2.4. Forward Translation

Two independent translators converted the original Italian version into Polish. Through the whole process of adaptation, one of the translators, who had a medical background, was involved. The second translator had a background outside of medicine.

### 2.5. Back-Translation and Expert Panel

A comparison of the original and two translated versions was achieved at this stage. The two translators, together with the authors, identified differences in translations and produced a combined version. Next, back-translation was carried out, where two independent translators, who were native in Italian, translated the Polish version into the original document’s language (Italian). Both translators were not familiar with the original version. At the following step, a commission group combined of translators, a psychologist, a specialist in orthopedics, and a statistician, assessed the translations. The so-called pre-final version was drafted as a result of consensus.

### 2.6. Pre-Testing, and Cognitive Interviewing and Development of Final Version

Thirty-four adolescents with AIS and two with SJK patients (who met the study eligibility requirements described above) were checked for the pre-final questionnaire’s comprehensiveness to ensure the adapted version was understandable. The participants were interviewed after completing the questionnaire in order to discuss their interpretation of each question and answer. The committee then reassessed this test’s outcome, and the final form of the questionnaire was created (Appendix A).

### 2.7. Statistical Analyses 

All statistical analyses were carried out using the Real Statistics Resource Pack software (Release 4.7) with a significance level of α < 0.05. The Shapiro–Wilk test for normality identified the data normally distributed; therefore, parametric tests were used. 

Furthermore, statistical analysis included descriptive statistics in calculating means ± standard deviations (SD) for a given question and the second level of analysis, which was comparative, concerning reliability, floor, and ceiling effect.

### 2.8. Reliability

The reliability was assessed using the two most important properties: consistency and stability. 

Internal consistency was assessed using Cronbach’s alpha coefficient. Cronbach’s alpha ranges from 0 (none of the items are correlated with one another) to 1 (reflects perfect internal consistency). Cronbach’s alpha of at least 0.70 was chosen to indicate adequate internal consistency [13].

Test–retest reliability was evaluated using the intraclass correlation coefficient (ICC_2.1_, CI = 95%), the most suitable and most commonly used reliability parameter for ordinal measures. ICC_2.1_ concerns the variation in the population (interindividual variation) divided by total variation, which is the interindividual variation plus the intraindividual variation (measurement error), expressed as a ratio between 0 and 1. The sum of scores obtained per each question provided by all patients during the first time and second time and the sum of total scores obtained per each patient were used, respectively, for test–retest analysis. A positive rating for reliability is when the ICC_2.1_ is at least 0.70 in a sample size of at least 50 patients [12]. To reduce the memory effect, there was a 7-day period between tests [12].

### 2.9. Measurement Error

The standard error of measurement (SEM) and minimal detectable change at the 90% level (MDC90) were employed. The sample included all patients (*n* = 58) who completed the ISYQOL-PL twice. The SEM was calculated as an element from the mean square error (MSE) from the analysis of variance, ANOVA. The MDC is the minimum change in a patient’s score that ensures the change is not the result of measurement error. The MDC90 was calculated using the formula: MDC = SEM × 1.65 × √2, where 1.65 is the z-value that reflects the 90% CI of no change, [14] and √2 indicates two measurements assessing change [14].

### 2.10. Floor and Ceiling Effects

Floor and ceiling effects were calculated and considered to be high if >15% of patients reported the worst or best status, respectively [12]. The distribution of results indicates the number (percentage) of patients with a minimum score (floor effect) and the number (percentage) of patients with a maximum score (ceiling effect).

## 3. Results

The duration of completing the questionnaire was ≤10 min in all patients.

The lowest, highest, and mean scores (%) obtained using the ISYQOL-PL are presented in Table 1.

Cronbach’s alpha coefficient was 0.8, which indicates satisfactory internal consistency (Table 2).

The test–retest study revealed high reliability, and the value of ICC_2.1_ for the entire group was high (0.90, CI ranged from 0.84 to 0.94) with SEM = 0.36 and MDC 90% CI = 0.84.

When completing the questionnaire for the first and the second time, there were no floor or ceiling effects for the ISYQOL-PL overall score. Cronbach’s alpha, mean (%), and SD, floor and ceiling effects for each ISYQOL-PL domain are presented in Table 3.

## 4. Discussion

This study presents a Polish adaptation of the ISYQOL, a new measure of HRQoL in adolescents with spinal deformities.

### 4.1. Statistical Relevance

Cronbach’s alpha is considered a proper method for estimating multi-item scales’ reliability, estimating internal consistency expressing the number of items and their average correlation [3]. Cronbach’s alpha should be greater than 0.70 to prove good reliability [13]. The ISYQOL-PL had a good value of Cronbach’s alpha coefficient (0.8), exceeding the minimum recommended value of 0.70 and indicating satisfactory internal consistency as a factor of satisfactory reliability of the ISYQOL-PL. The Cronbach’s alpha score achieved by authors of the original version, Caronni et al. [7], was 0.83. The English version of the ISYQOL questionnaire had the value of Cronbach’s alpha of 0.79 to 0.84 [9], indicating good internal consistency [1]. The Spanish version of the ISYQOL showed an acceptable internal consistency (0.77 Cronbach’s alpha) [10] and is considered a reliable and valid tool to measure HRQoL admitted to Spanish-language-speaking adolescents with spinal deformity during conservative treatment (with or without a brace). There was neither floor nor ceiling effect for the ISYQOL-PL overall score. Sánchez-Raya et al. [10] reported similar results.

### 4.2. Clinical Relevance

The conservative treatment of adolescents with spinal deformities without or with a brace can significantly impact patients’ subjective perception and social well-being and negatively affect their HRQoL [1]. Moreover, Wang et al. [15], in a recent review analyzing the impact of bracing on AIS patient’s quality of life, indicated that self-image, mental health and vitality are the three most frequently reported affected domains. 

The treatment’s effectiveness has been demonstrated to depend on the patients’ treatment compliance [16,17,18] and the quality of brace usage [18]. Additionally, the brace’s impact on the self and body image is reported as a contributing factor for stress. The stress level determines compliance and can be assessed using the BSSQ questionnaire (BSSQ—Deformity, BSSQ—Brace). Kotwicki et al. [19] noticed that the BSSQ is a helpful method for determining the level of stress during conservative treatment for AIS patients. Lin et al. [20] found that female adolescents with idiopathic scoliosis who had undergone bracing were more vulnerable to depressive psychological status. 

However, the international scientific Society on Scoliosis Orthopaedic and Rehabilitation Treatment (SOSORT) recommends the overall evaluation of adolescents with spinal deformities using HRQoL questionnaires [21]. The ISYQOL questionnaire, developed with Rasch analysis, fully complies with a fundamental measure (additivity, generalizability, and unidimensionality) [8]. The ISYQOL items were developed based on the concerns expressed by the patient and the clinician’s input. Moreover, Caronni et al. showed high validity using HRQoL in adolescents with spinal deformities and better known-groups validity than the very often used SRS22 questionnaire [8]. Next of the ISYQOL questionnaire’s strengths is the possibility to compare the measurement of the HRQoL in patients with AIS and SJK and patients wearing or not wearing a brace [7]. 

Our results indicate that an ISYQOL-PL questionnaire is a reliable tool for evaluating the HRQoL of adolescent patients with spinal deformities. 

As a limitation, the ISYQOL-PL presented a lower sample than the Italian sample (402 Italian patients versus 58 Polish). Additionally, our group consisted of six boys and two patients with SJK, so it was impossible to analyze the per-gender and per-spinal deformities differences. Investigating this specific aspect at the larger sample size is under our ongoing study. Finally, in the future, the Polish version of the ISYQOL questionnaire could be tested in patients who had surgical treatment and adult patients, as recommended by Caronni et al. [8].

## 5. Conclusions

The ISYQOL-PL is a brief and practical tool for quantifying HRQoL in adolescents with a spine deformity. Filling in the questionnaire takes less than 10 minutes to be completed. The ISYQOL-PL questionnaire is reliable and can be used in adolescents with spinal deformities.

## Figures and Tables

**Table 1 jcm-10-02081-t001:** Distribution of minimal, maximal, and mean total scores (%) of ISYQOL-PL.

Questionnaire	N	Min	Max	Mean	SD
ISYQOL-PL first trial	58	32.96%	62.83%	49.05%	3.15
ISYQOL-PL second trial	58	32.96%	66.29%	49.05%	3.15

**Table 2 jcm-10-02081-t002:** The Cronbach’s alpha value coefficient of the ISYQOL-PL compared to the ISYQOL (original).

ISYQOL-PL	ISYQOL (Original)
Cronbach’s alpha	Cronbach’s alpha
0.8	0.83

**Table 3 jcm-10-02081-t003:** Cronbach’s alpha, mean (%), and SD, floor and ceiling effects for each ISYQOL-PL domain.

ISYQOL-PL DOMAIN	Number of Items	Cronbach’s Alpha	Mean	SD	Floor Effect	Ceiling Effect
SPINE HEALTH	13	0.79	56.88%	3.32	0 (0.0%)	0 (0.0%)
BRACE	7	0.77	68.21%	4	0 (0.0%)	1 (1.7%)

## Data Availability

The data presented in this study are available on request from the corresponding author.

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
