# Peer review of "Polish Adaptation of the Italian Spine Youth Quality of Life Questionnaire"

_jcm, 2021, doi:10.3390/jcm10102081_

Round 1
Reviewer 1 Report
The study was very well done. The authors kept it simple and conducted good process. It would be significant to the patient population (adolescent) with primary spine deformity.
Summary:
Polish adaptation of the Italian Spine Youth Quality of Life 2 questionnaire
Introduction: background information clearly outlined and aim of the study stated to be: The paper aims to carry on cultural adaptation 50 of the ISYQOL questionnaire into Polish (ISYQOL-PL).
Would insert the priori hypothesis closer to the introduction:
“the ISYQOL-PL questionnaire ver-76 sion is reliable and appropriate for adolescents with spinal deformity”
Study design
Methods: REB approval was obtained; subjects identified and accounted for; inclusion/exclusion criteria stated; development of the questionnaire was clearly outlined; developed and underwent a content review; the analyses seemed reasonable;
Results were clearly stated and back-up with the analyses performed
Discussion commented on the similarities of results from the Spanish and English versions to the current Polish version; clinical relevance was identified for this population of adolescent patients; Limitations were outlined (low sample size and more girls than boys with the few boys presenting with SJK so no gender comparisons were possible)
Conclusions were well written and concise indicating that the ISYQOL-PL is a brief and practical tool for quantifying HRQoL in adolescents 258 with spine deformity. Filling in the questionnaire takes less than 10 min to be completed; it is reliable and used for adolescents with spinal deformity.
Author Response
We thank the reviewer for the valuable work and suggestions.
Would insert the priori hypothesis closer to the introduction:“the ISYQOL-PL questionnaire ver-76 sion is reliable and appropriate for adolescents with spinal deformity”
Response
We moved the study hypothesis closer to the Introduction section, lines: 55-56.
Reviewer 2 Report
This is a well executed study in translating the ISYQOL to Polish. The authors have done a great job with a proper methodology. I have some questions and comments that might further improve this paper.
- It has not been stated clearly how much childrens parents were allowed to assist in answering the questionnaire. It would be good for readers to know the extent of parental infleunce on data provided. Secondly, it seems that the children were obviously monitored at the clinic when answering the questionnaire. Do the authors feel that the children would answer the questionnaire differently if they were in their own domestic context without the observation of the research personnel?
- Data on number of individuals invited initially to participate and response rate would be good to present and would further strengthen the study. Was this a representative sample? Like the authors have stated in the limitations section; it was a fairly small sample and especially for SJK where only two individuals participated.
Author Response
We thank the reviewer for the valuable work and suggestions.
- It has not been stated clearly how much childrens parents were allowed to assist in answering the questionnaire. It would be good for readers to know the extent of parental infleunce on data provided. Secondly, it seems that the children were obviously monitored at the clinic when answering the questionnaire. Do the authors feel that the children would answer the questionnaire differently if they were in their own domestic context without the observation of the research personnel?
- Data on number of individuals invited initially to participate and response rate would be good to present and would further strengthen the study. Was this a representative sample? Like the authors have stated in the limitations section; it was a fairly small sample and especially for SJK where only two individuals participated
Response
Thanks for the questions.
Answer for questions 1 and 2:
We added to the Materials and Methods section the following:
Lines:74-76 "All the invited adolescents selected through the inclusion criteria participated in the study. The participants were left to fill in the questionnaire alone, being in a separate space, to minimize any influence from parents or medical staff."
The study aimed to establish if the ISYQOL-PL questionnaire could be appropriately applied to Polish adolescents with spinal deformities. As we stated in the limitations section sample size was small but large enough, following the literature recommendations [1,2], to study the ISYQOL-PL questionnaire reliability properly.
The study was not aimed to describe which is the representative HRQoL level for Polish adolescents with spinal deformities population. For such research, a larger representative sample of Polish adolescents with spinal deformities population is needed. Moreover, the study did not aim to evaluate different environmental factors that could influence the ISYQOL-PL questionnaire filling in process. Both above issues could be a matter for future study.
- Terwee, C.B.; Bot, S.D.M.; de Boer, M.R.; van der Windt, D.A.W.M.; Knol, D.L.; Dekker, J.; Bouter, L.M.; de Vet, H.C.W. Quality Criteria Were Proposed for Measurement Properties of Health Status Questionnaires. Journal of Clinical Epidemiology 2007, 60, 34–42, doi:10.1016/j.jclinepi.2006.03.012.
- Tsang, S.; Royse, C.; Terkawi, A. Guidelines for Developing, Translating, and Validating a Questionnaire in Perioperative and Pain Medicine. Saudi J Anaesth 2017, 11, 80, doi:10.4103/sja.SJA_203_17.
This manuscript is a resubmission of an earlier submission. The following is a list of the peer review reports and author responses from that submission.